# Long-Term Changes to the Microbiome, Blood Lipid Profiles and IL-6 in Female and Male Swedish Patients in Response to Bariatric Roux-en-Y Gastric Bypass

**DOI:** 10.3390/nu16040498

**Published:** 2024-02-09

**Authors:** Olena Prykhodko, Stephen Burleigh, Magnus Campanello, Britt-Marie Iresjö, Thomas Zilling, Åsa Ljungh, Ulrika Smedh, Frida Fåk Hållenius

**Affiliations:** 1Division of Food and Pharma, Department of Process and Life Science Engineering, Faculty of Engineering, Lund University, 221 00 Lund, Sweden; stephen.burleigh@food.lth.se (S.B.); frida.hallenius@food.lth.se (F.F.H.); 2Department of Surgery, Halland Regional Hospital Varberg, 432 81 Varberg, Sweden; magnus.campanello@hotmail.com (M.C.); thomas.zilling@slf.se (T.Z.); 3Department of Surgery, Institute of Clinical Sciences, Sahlgrenska Academy at the University of Gothenburg, 413 45 Gothenburg, Sweden; britt-marie.iresjo@surgery.gu.se (B.-M.I.);; 4Medical Faculty, Lund University, 221 00 Lund, Sweden; asahelena.ljungh@gmail.com

**Keywords:** obesity, weight loss surgery, gut microbiota, blood cholesterol, low-density lipoprotein, high-density lipoprotein, *Prevotella*

## Abstract

Lipid metabolism dysregulation is a critical factor contributing to obesity. To counteract obesity-associated disorders, bariatric surgery is implemented as a very effective method. However, surgery such as Roux-en-Y gastric bypass (RYGB) is irreversible, resulting in life-long changes to the digestive tract. The aim of the present study was to elucidate changes in the fecal microbiota before and after RYGB in relation to blood lipid profiles and proinflammatory IL-6. Here, we studied the long-term effects, up to six years after the RYGB procedure, on 15 patients’ gut microbiomes and their post-surgery well-being, emphasizing the biological sex of the patients. The results showed improved health among the patients after surgery, which coincided with weight loss and improved lipid metabolism. Health changes were associated with decreased inflammation and significant alterations in the gut microbiome after surgery that differed between females and males. The Actinobacteriota phylum decreased in females and increased in males. Overall increases in the genera *Prevotella*, *Paraprevotella*, *Gemella*, *Streptococcus*, and *Veillonella_A*, and decreases in *Bacteroides_H*, *Anaerostipes*, *Lachnoclostridium_B*, *Hydrogeniiclostridium*, *Lawsonibacter*, *Paludicola*, and *Rothia* were observed. In conclusion, our findings indicate that there were long-term changes in the gut microbiota after RYGB, and shifts in the microbial taxa appeared to differ depending on sex, which should be investigated further in a larger cohort.

## 1. Introduction

Obesity is a disorder, mainly related to lipid metabolism, which has reached pandemic levels. According to Martinez et al., obesity afflicts 36.5% of the US population and 600 million individuals worldwide [1]. Obesity predisposes humans to multiple comorbidities, including type 2 diabetes, various forms of cancer, hypertension, hyperlipidemia, sleep apnea, heart disease, increased risk of stroke, gastroesophageal reflux disease, and osteoarthritis, and it is also linked to low socioeconomic status, unemployment, and reduced productivity [2,3]. Morbid obesity is often attributed to low physical activity in combination with too much energy-dense food [4].

Roux-en-Y gastric bypass (RYGB) is an effective method to treat obesity, and significant loss of body fat after this surgery has been observed in controlled studies [5,6]. Laparoscopic RYGB can be performed with different anastomotic techniques to create the gastrojejunostomy, and the procedure permanently alters the body’s normal digestive processes. Measuring successful outcomes of the surgery, apart from weight loss, is achieved using a variety of physical and blood-based biomarkers such as lipid profiles, glucose, and also hormones such as ghrelin [7]. Quality of life (QoL) assessments such as the Moorehead–Ardelt questionnaire [8] are also used to help judge the outcome of the RYGB. It is worth mentioning that a gender-based disparity in the utilization of weight loss surgery such as RYGB among eligible patients has been reported [9].

The gut microbiome plays a central role in human health, both as a cause of disease and as a powerful force in disease prevention and nutritional well-being [10,11,12]. Studies of the gut microbiome have become more common in recent years as costs have dramatically reduced and sampling technologies have improved [13]. Numerous bacteria are involved in various beneficial metabolic processes and immunity, although the exact mechanisms of action are poorly understood [14,15]. Bacteria of special interest include genera which are highly involved in the production of butyrate, a short-chain fatty acid shown to improve vascular and gut barrier functions and decrease inflammation [16,17], and lactic acid-producing bacteria such as *Lactobacillus* and *Bifidobacterium*, which can be involved in improving digestive tract health [18].

There is compelling evidence of an essential role for the gut microbiota in the development and perpetuation of obesity [19]. Obese patients can suffer from dysbiosis of the microbiota characterized by decreased diversity in the microbial community and by an increased Firmicutes–Bacteroidetes ratio [20]. Moreover, the gut microbiota composition differs between obese men and women [21]. The gut microbiota composition changes after bariatric surgery, but the mechanism behind this change is unknown [22,23]. One explanation that has been proposed is that the gut microbiota of obese individuals adapts to extracting more energy from the diet than the microbiota of lean individuals [24].

A better understanding of how the gut microbiota composition is influenced by bariatric surgery with a long-term perspective and whether there are any associations with well-being factors such as improvement of lipid metabolism, decreased inflammation and quality of life is needed. Therefore, the purpose of the present study was to evaluate the long-term changes, up to six years after gastric bypass surgery, to the gut microbiota of patients and to relate these changes to biomarkers and surveys such as the Moorehead–Ardelt Quality of Life Questionnaire, emphasizing sex-related differences.

## 2. Materials and Methods

### 2.1. Ethical Statement

The study was approved by the Regional Ethical Review Board in Lund, Sweden (Reference number 354/2008). All study participants gave written informed consent in accordance with the Declaration of Helsinki and its later amendments.

### 2.2. Patients

Inclusion criteria for the present study were age over eighteen years, several previous weight loss attempts, non-smokers, and subjects should not previously have undergone any form of weight loss surgery. Initially, twenty obese patients with BMI 37.9 ± 1.1 (range 31–50) scheduled for bariatric surgery were recruited, but five dropped out due to unwillingness to participate, leaving fifteen patients (eleven females and four males) ranging between 19 and 61 years of age to complete the study (average age 41). The BMI prior to surgery of these patients was 37.6 ± 1.1 (range 31–44). It is worth mentioning that the eligibility criteria for the patients included BMI > 35, but 4 patients were accepted with a lower BMI (31, 32, 33, 34) due to comorbidities in the form of type 2 diabetes (3), hypertension (4), sleep apnea (1), depression (3), and osteoarthritis (4). The exclusion criteria were severe eating disorders, substance abuse, and/or ongoing episodes of mental illness.

### 2.3. Surgical Techniques

Two different techniques were randomly assigned independently of age, BMI, biological gender, or health conditions. Roux-en-Y gastric bypass was accomplished using an antecolic antegastric laparoscopic approach using five ports [25]. Briefly, the surgery started with the creation of a proximal gastric pouch by complete division of the stomach using three cartridges of a blue 45 mm linear stapler (Ethicon Endo-Surgery, LLC., Guaynabo, PR, USA). The line of gastric division was started 3–4 cm distal to the right gastroesophageal junction, perpendicular to the lesser curve, and then continued vertically to the left gastroesophageal junction. Before the anastomosis was created, the omentum was divided systematically to relieve tension on the anastomosis.

For the circular stapled anastomosis, the EEA TM 21 Or Vil (Covidien, Mansfield, MA, USA) was used. The cartridge was passed by transoral delivery into the created small gastric pouch (15–20 mL). After creating a small opening in the suture line of the pouch using ultrasonic-activated scissors (Ultra Cision Harmonic Scalpel^®^, Ethicon Endo-Surgery, LLC., Guaynabo, PR, USA), the end of the antecolic Roux-Loop was opened after the bowel previously had been divided 150 cm from the Ligamentum of Treitz. The EEA stapler was introduced into the small bowel and connected to the cartridge in the gastric pouch, and an end-to-side circular anastomosis was constructed.

For the linear anastomosis, the small bowel was brought up as a 150 cm antecolic loop from the Ligamentum of Treitz, and a gastro-jejunal anastomosis was performed with the Endopath ETS articulating Linear cutter 45 mm reload GR 45B blue cartridge (Ethicon Endo-Surgery, LLC., Guaynabo, PR, USA). The remaining opening in the anastomosis was closed with a 3–0 polyglactin running suture.

Closure of the mesenteric defects was not performed in this study. The anastomosis of the gastrojejunostomy was assessed by endoscopy perioperatively and measured from zero to six years after surgery.

### 2.4. Material Sampling and Biomarkers

To assess surgical outcome and its relationship with the patient microbiome, several biomarkers were studied. Parameters such as body weight (BW, kg), body mass index (BMI, kg/m^2^), as well as quality of life (QoL) assessments such as the Moorehead–Ardelt questionnaire (Moore) [8] and the Physical Composite Score (PCS) [26] were investigated before and up to six years post-surgery, with a focus on self-esteem, physical well-being, social relationships, work, and sexuality.

Biological material such as blood and stool samples was collected before the surgery and up to six years post-surgery. See Appendix A.

All patients fasted for a minimum of 6 h before blood collection. Blood samples for albumin (Alb), iron (Fe), cholesterol (Chol), triglycerides (Tg), high-density lipoprotein (HDL), low-density lipoprotein (LDL), and glucose (Glu) were collected from an arm vein into 5 mL Lithium-heparin BD vacutainers, while blood for glycated hemoglobin (HbA1c) was collected into 5 mL EDTA vacutainers according to clinical standards and analyzed on the day of collection at the clinical chemistry laboratory at Varberg County Hospital. The methods used were immunoturbidimetric analysis (Alb), colorimetric (Fe), enzymatic colorimetric (Chol, HDL, LDL, Tg), photometric (Glu), and turbidimetric inhibition immunoassay for HbA1c.

Blood samples for ghrelin and interleukin 6 (IL-6) were collected in cooled sodium heparin tubes with the addition of 500 KIE aprotinin/mL blood (Trasylol^®^ injection fluid 10,000 KIE/mL). Blood samples were kept on ice and centrifuged within 30 min of blood collection at 2500× *g* for 10 min at +4 °C. Plasma was collected, aliquoted, and stored at −70 °C until analysis. Total plasma ghrelin concentrations were determined with a Human Total Ghrelin Elisa kit (EZGRT-89 K, Merck Milipore, kGaA, Darmstadt, Germany) according to the manufacturer’s instructions. The assay detects both active and des-octanoyl forms of ghrelin. Sample aliquots for ghrelin measurements were acidified with HCL (final concentration 0.05 M) and analyses were performed in duplicate. Interleukin-6 concentrations were determined with the Immulite method at the Department of Clinical Immunology, Sahlgrenska University hospital, Gothenburg. The lower limit of quantification (LLOQ) for the IL-6 assay was 2 pg/mL.

Stool samples were collected by patients themselves in provided and labeled sterile tubes (Sarstedt, Nümbrecht, Germany), transported to the research unit within 30 min, and stored at −80 °C until analysis.

### 2.5. DNA Extraction, PCR Amplification, and 16S rRNA Gene Sequencing

DNA from 50 to 100 mg of fecal samples was extracted using the QIAamp DNA Stool Mini Kit (Qiagen, Hilden, Germany) according to the manufacturer’s instructions. Measurement of DNA concentration was performed using Qubit 2.0 Fluorometer (Thermo Fisher Scientific, Waltham, MA, USA).

The hypervariable V4 region of the 16S rRNA genes was amplified using primers 515F 5′-GTGCCAGCMGCCGCGGTAA-3′ and 806R 5′-GGACTACHVGGGTWTCTAAT-3′ for forward and reverse reads, respectively [27]. Paired-end sequencing with a read length of 2 × 250 bp was carried out on a Miseq Instrument (Illumina, San Diego, CA, USA) using the Miseq reagent kit v2 (Illumina, San Diego, CA, USA). As an internal control, 5% of PhiX was added to the amplicon pool. Illumina sequencing adaptors were trimmed off during the generation of FASTQ files and reads that did not match any barcodes were discarded.

### 2.6. Bioinformatics and Sequence Analysis

Sequence data were analyzed with the open-source bioinformatics pipeline Quantitative Insights Into Microbial Ecology, QIIME v2 following the standard pipeline [28]. Sequences were removed when lengths were <220 nucleotides, >290 nucleotides or when the quality score fell below 25 [29]. The sequences were normalized by rarefaction (depth of 32,071) and grouped into operational taxonomic units (OTUs) at a minimum of 97% similarity using the closed reference method based on the Greengenes database (v. 13.8) [30].

A QIIME-based Permanova (using the pseudo-F statistical test and 999 permutations) was used to test for overall differences between the microbiomes of the pre- and post-operation patients. The Pearson correlations table was generated using Python3 (Python Software Foundation, 2020) and the Python library “Matplotlib”. The PCA biplot was created using Matlab (v. R2022a).

### 2.7. Statistics and Calculations

Calculations, statistical analyses, and graphs were completed using GraphPad Prism, Version 10.1.1. A paired *t*-test was performed in order to compare variables from individuals in pre- and post-surgery groups, while a 2-way ANOVA test was performed with sources of variation such as sex, time (pre- and post-) and subject using post hoc Šídák’s multiple comparisons test. Significant (alpha 0.05) differences are indicated as ****, ***, ** and * where *p* < 0.0001, *p* < 0.001, *p* < 0.01, and *p* < 0.05, respectively, while the symbol # shows trend where *p* < 0.07. Results per each parameter are presented as mean ± SEM.

## 3. Results

### 3.1. Surgery Outcome

#### 3.1.1. Effect on Body Weight and Quality of Life

Surgery resulted in significant fat loss for both female and male patients, independently of age, which manifested as reduced body weight (BW) and BMI (Figure 1A,B). The body percentage reduction in weight achieved by the patients during the studied period was 23.97 ± 2.93 (range 7.6–30%) for women and 22.56 ± 5.24 (range 11.45–34.35%) for men, and it was associated with improved quality of life, including improved self-esteem, which was significantly higher for female patients (Figure 1C,D). Interestingly, the Moore score was much higher in male patients (1.83 ± 0.19) compared to females (0.42 ± 0.34), even before the bariatric surgery.

#### 3.1.2. Blood Parameters

The surgery significantly affected blood lipid profiles (Table 1), resulting in improved lipid metabolism for the whole cohort, which manifested as decreased blood cholesterol (from 5.53 ± 0.35 to 4.33 ± 0.22 mmol/L, *p* < 0.001), LDL (from 3.57 ± 0.27 to 2.87 ± 0.21 mmol/L, *p* < 0.01) and LDL/HDL ratio (from 3.31 ± 0.32 to 1.95 ± 0.17, *p* < 0.0001), and increased HDL levels (from 1.15 ± 0.08 to 1.55 ± 0.11 mmol/L, *p* < 0.001), while triglyceride levels had only a tendency to decrease. Blood glucose was significantly lower after surgery in both female and male patients (from 5.59 ± 0.31 to 5.03 ± 0.26 mmol/L, *p* < 0.01), while glycated hemoglobin levels as well as albumin and iron levels were unchanged after surgery. The appetite regulation-related hormone, ghrelin, was significantly increased after surgery for the studied cohort, particularly in female patients. Inflammatory IL-6 decreased after surgery (from 3.69 ± 0.74 to 2.48 ± 0.65 pg/mL, *p* < 0.05), and it was specifically significantly lower for female patients (*p* < 0.01).

#### 3.1.3. Gut Microbiome

Sequence analysis of the 16S rRNA genes resulted in 6420 OTUs assigned to L2-L6 taxonomic levels (from phylum to genus) of gut bacteria. There were significant differences in the taxonomic profiles of the pre- and post-operation states. The phylum level (Figure 2) showed overall decreased Bacteroidota from 59.8% to 54.7% and increased Firmicutes, which was significant for male patients for Firmicutes A (from 29.7 to 35.3%) and D (from 0.97 to 1.55%), while female patients showed a significant increase only for Firmicutes C (from 2.59 to 5.08%) (Figure 2A). The ratio of Firmicutes to Bacteroidota (F/B) resulted in a significant increase only for male patients after surgery (Figure 2B). There were no significant differences observed for the other phyla before and after surgery, except for the Actinobacteriota phylum, which significantly increased in male patients (from 0.31 to 0.91%) and decreased in female patients (from 1.6 to 0.64%) (Figure 2C). Increases in the genera *Prevotella*, *Paraprevotella*, *Gemella*, *Streptococcus*, and *Veillonella_A*, and decreases in *Bacteroides_H*, *Anaerostipes*, *Lachnoclostridium_B*, *Hydrogeniiclostridium*, *Lawsonibacter*, *Paludicola*, and *Rothia*, were observed for the total cohort of patients (*p* < 0.05). However, sex-related changes included increased abundance of *Faecalibacterium* and *Roseburia* and decreased *Bifidobacterium* in females after surgery, while in male patients increased abundance of *Collinsella*, a genus that belongs to the *Corinobacteriaceae* family in the Actinobacteriota phylum, was observed in the present study (Figure 2D–O). The visual abundance of all bacteria at the phylum and genus level per group and in individual patients before and after surgery is presented in Appendix A.

### 3.2. Relation of Microbial Taxa to Biomarkers

Multivariate data analysis revealed a complex set of interactions between pre- and post-operation groups, QoL assessments, biomarkers, and the gut microbiome (Figure 3A). For instance, *Streptococcus* was strongly correlated with the post-surgery group, while cholesterol and LDL levels were negatively correlated with the *Streptococcaceae* family and positively correlated with *Paludicola*. Blood glucose was strongly associated with BW, BMI, as well as with the *Burkholderiaceae* family, while it was negatively correlated with *Phocaeicola*. Inflammatory marker IL-6 showed an association with *Blautia* and *Dorea* and was negatively correlated with the *Gemella* genus. As seen in the principal component analysis (PCA) biplot (Figure 3B) BW, BMI, cholesterol, LDL, and fasting glucose were associated with the pre-operation group (Y0), while HDL, ghrelin, Moore, and PCS were correlated with the post-operation group (Y1-6). Furthermore, there was an association between the post-operation group and the *Prevotella* genus and HDL blood levels. Additionally, the two surgical anastomosis techniques applied in the present study had no strong association with either the pre- and post-operation groups or any of the patient biomarkers and therefore will not be further discussed.

## 4. Discussion

This study was carried out to identify changes to the human microbiome after bariatric Roux-en-Y gastric bypass surgery in relation to changes in blood lipid profiles and IL-6 in female and male patients. Given that this study focused on the post-surgery period and was based on samples from patients studied from one to six years after surgery, our results can be considered part of a long-term study, with the findings likely to be associated with not only the direct effects of the surgery, but also with possible changes in the patients’ lifestyles and diets associated with long-term post-surgery weight loss. Data regarding the individual changes in lifestyle and nutrient intake during the pre- and post-surgical period are not available, which is a limitation of the present study.

As evidenced by the reduced BW, BMI, cholesterol, LDL, and fasting glucose, along with a consequent increase in HDL and improved Moore and PCS scores in the post-surgery group, we concluded that the surgical interventions were beneficial and lasting with regard to patient health. Interestingly, female patients showed very low Moore scores before the surgery compared to male patients, and they also had a more dramatic increase in their quality-of-life score after surgery. The overall beneficial health effect during the post-operative period (from one up to six years) was associated with decreased inflammation and alterations in the microbiomes of the patients that might be due to irreversible changes in their gastrointestinal anatomy, as well as to altered diet and/or changes in exercise regimes.

### RYGB Effects on Gut Microbiota

The post-surgery response in female patients was not reflected in F/B ratio changes. However, it is clear that changes in Firmicutes and Bacteroidota abundance still plays a key role in the etiology of obesity, as reported for obese populations [31]. Interestingly, observed changes in the microbiota composition of patients after bariatric surgery were not shifting towards “lean type” microbiota, despite the dramatic weight loss and normalization of glucose and lipid metabolism. For instance, increased Firmicutes was observed for the whole cohort, while the decrease in Bacteroidota and *Bacteroides_H* was significant in male and female patients, respectively. Furthermore, decreased abundance of propionate- and acetate-producing bacteria such as *Anaerostipes* and butyrate producer *Lawsonibacter* might relate to dietary changes in, for instance, dairy and coffee consumption, respectively [32,33]. However, increases in some beneficial bacteria were also seen; for instance, for *Roseburia* and *Faecalibacterium*, known butyrate producers which are also considered to be anti-inflammatory and play an important role in intestinal health by improving intestinal barrier function [34]. The increased *Prevotella* that was highly associated with the post-surgery group might be explained by changed dietary preferences among patients to more fiber-rich foods, which is also known to be involved in normalization of glucose metabolism [35]. Furthermore, increased abundance of both *Prevotella* and *Paraprevotella* might also be associated with protein metabolism and digestion of dietary proteins [36], since increased maldigestion and malnutrition due to bariatric surgery has been reported [37]. Interestingly, increased *Collinsella* abundance, observed only in male patients, might indicate different dietary preferences between male and female subjects after surgery, since increases in *Collinsella* have been shown to be associated with insufficient intake of dietary fibers in obese individuals [38].

Post-surgery taxonomic changes in the microbiome also included bacteria not usually considered beneficial to humans, including an increase in *Veillonella* and *Streptococcus*. For instance, *Veillonella* has been implicated in periodontal disease [39] and has also been reported to increase in another cohort of obese diabetic patients after one year post-RYGB surgery [40]. In RYGB, there is only a minimal pocket of stomach left which is exposed to saliva and food, while the main part of the stomach volume and whole duodenum are left disconnected and unexposed to oral contents. In addition, the oro-intestinal passage time is significantly more rapid after the surgery. Thus, it is possible, if not likely, that oral bacteria may have a greater chance of surviving the acidic conditions of the stomach and colonizing the intestines. Therefore, some of the long-term effects of bariatric surgery on the gut microbiota composition may not be associated with an improved colonic environment.

## 5. Conclusions

Improved lipid and glucose metabolism and decreased inflammation after bariatric Roux-en-Y gastric bypass were found in Swedish patients followed up to six years post-surgery. Permanent surgical alteration of gastrointestinal anatomy with secondary weight normalization caused clear and long-term changes in the gut microbiota, even in a relatively small cohort. Notably, the gut abundance of *Prevotella* increased significantly post-surgery, and the level of the inflammatory marker IL-6 was positively correlated with *Blautia* and *Dorea* and negatively correlated with *Gemella*. Since the long-term health impact of this is currently unknown, further studies in a larger patient cohort are merited. In addition, differences in dietary preferences and lifestyle between female and male patients should be included to better understand patterns of microbiota composition alterations as an informative marker to monitor and/or predict health outcomes of patients after bariatric surgery with a long-term perspective.

## 6. Limitation of Study

Low sample size due to patient drop-out is a limitation of this long-term study.

## Figures and Tables

**Figure 1 nutrients-16-00498-f001:**
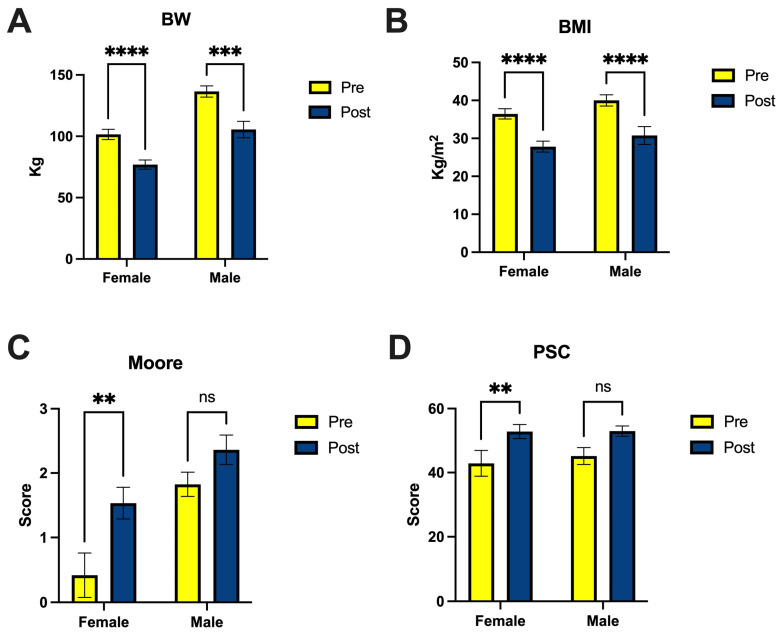
The surgery outcome for female (*n* = 11) and male (*n* = 4) patients resulted in changes in body weight (**A**), BMI (**B**) and quality of life, assessed by Moore (**C**) and PSC (**D**) questionnaires. BW—body weight; BMI—body mass index; Moore—Moorehead–Ardelt; PSC—Physical Composite Score. Results presented as mean ± SEM; significance indicated as ** *p* < 0.01, *** *p* < 0.001, **** *p* < 0.0001; ns, not significant.

**Figure 2 nutrients-16-00498-f002:**
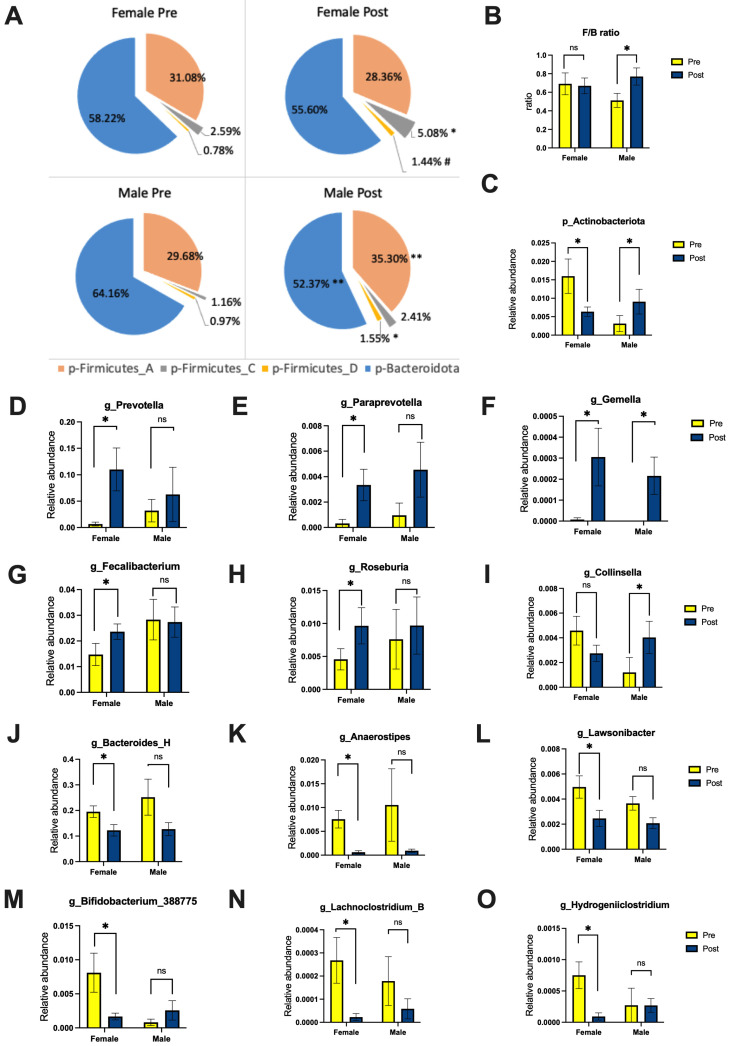
The change in gut microbiome in female (*n* = 11) and male (*n* = 4) patients before and after surgery on the phylum (**A**–**C**) and genus levels (**D**–**O**). Results are presented as mean ± SEM; significance is indicated as * *p* < 0.05, ** *p* < 0.01, while trend is indicated as # *p* < 0.07; ns—not significant.

**Figure 3 nutrients-16-00498-f003:**
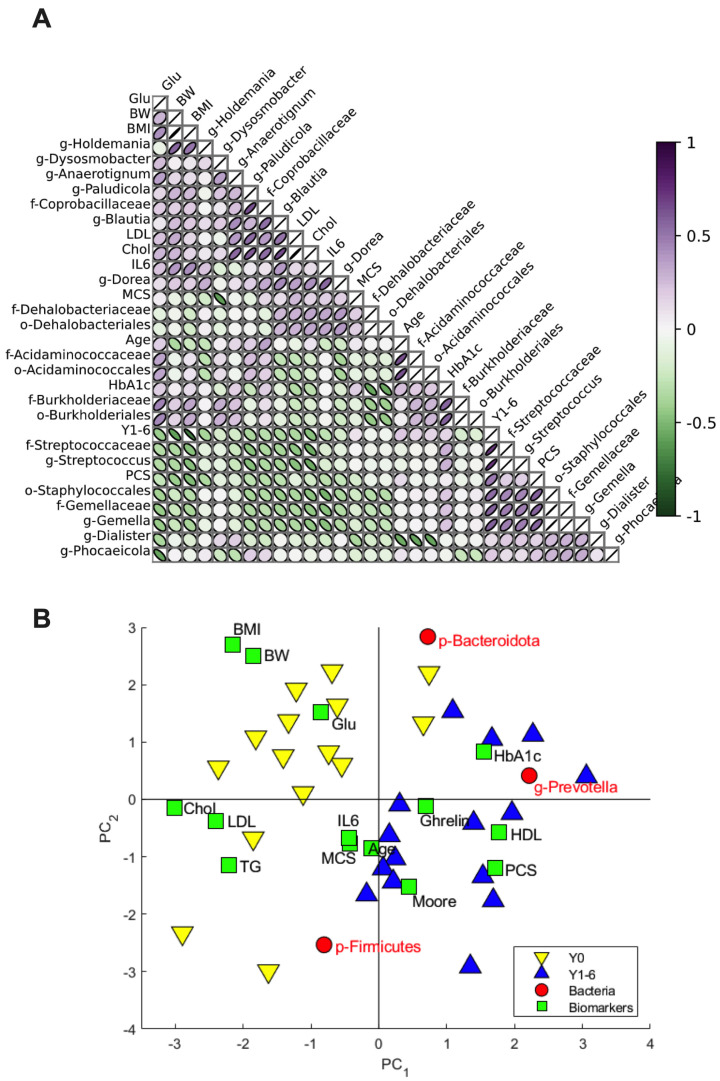
Pearson’s correlation table (**A**) showing positive (purple) and negative (green) relations between a selection of the bacteria and biomarkers (30 XY pair). The darker and more linear the symbols are, the greater the correlation. White circles represent no correlation; PCA biplot of the loading variables (**B**) (selected bacteria (red circles) and biomarkers (green squares)) and scores (patients) are colored according to pre- (yellow inverted triangles, *n* = 15) and post-surgery (blue triangles, *n* = 15) status.

**Table 1 nutrients-16-00498-t001:** Blood markers of patients before the weight loss surgery and at follow-up (up to 6 years later).

	Before Surgery			After Surgery		
Blood Markers, Units	Female	Male	Total	Female	Male	Total
Alb, mmol/L	38.27 ± 0.98	40.00 ± 2.68	38.73 ± 0.98	37.00 ± 1.018	39.50 ± 2.67	37.67 ± 1.02
Fe, mmol/L	14.82 ± 1.60	16.75 ± 3.09	15.33 ± 1.39	14.46 ± 1.19	23.25 ± 2.93	16.80 ± 1.52
Lipid metabolism:						
Chol, mmol/L	5.17 ± 0.35	6.53 ± 0.71	5.53 ± 0.35	4.10 ± 0.25 **	4.95 ± 0.30 *	4.33 ± 0.22 ***
Tg, mmol/L	1.17 ± 0.21	1.25 ± 0.22	1.19 ± 0.16	0.81 ± 0.11 #	1.30 ± 0.23	0.94 ± 0.11 #
HDL, mmol/L	1.17 ± 0.10	1.08 ± 0.10	1.15 ± 0.08	1.55 ± 0.11 **	1.55 ± 0.30 *	1.55 ± 0.11 ***
LDL, mmol/L	3.16 ± 0.19	4.68 ± 0.61	3.57 ± 0.27	2.68 ± 0.25	3.40 ± 0.29 **	2.87 ± 0.21 **
LDL/HDL, ratio	2.90 ± 0.31	4.42 ± 0.60	3.31 ± 0.32	1.80 ± 0.18 ***	2.36 ± 0.33 ***	1.95 ± 0.17 ****
Glucose metabolism:						
Glu, mmol/L	5.67 ± 0.42	5.35 ± 1.55	5.59 ± 0.31	5.15 ± 0.30 *	4.70 ± 0.18 *	5.03 ± 0.26 **
HbA1c, mmol/mol	38.73 ± 19.10	30.28 ± 15.13	38.00 ± 1.94	39.7 ± 11.99	34.20 ± 2.86	37.87 ± 2.61
Appetite regulation:						
Ghrelin, pg/mL	623.36 ± 106.23	665.5 ± 49.59	634.60 ± 77.95	801.82 ± 124.17 **	809.75 ± 42.89	803.93 ± 90.45 ***
Inflammation:						
IL-6, pg/mL	4.21 ± 0.89	2.89 ± 0.92	3.69 ± 0.74	2.43 ± 0.62 **	2.64 ± 1.24	2.48 ± 0.65 *

Results are presented as mean ± SEM for female (*n* = 11) and male patients (*n* = 4) as well as for total cohort (*n* = 15) comparison before the weight loss surgery and up to 6-year follow-ups. Significant changes in blood parameters before and after surgery are indicated as ****, ***, **, and * where *p* < 0.0001, *p* < 0.001, *p* < 0.01, and *p* < 0.05, respectively, while symbol # shows trend where *p* < 0.07. Alb, albumin; Fe, iron; Chol, cholesterol; Tg, triglycerides; HDL, high-density lipoprotein; LDL, low-density lipoprotein; Glu, glucose; HbA1c, glycated hemoglobin; IL-6, interleukin 6.

## Data Availability

The microbiome and associated biomarker data are available through the Sequence Read Archive (SRA) at the National Center for Biotechnology Information (NCBI). The BioProject identification number is PRJNA808195.

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
