# Peer review of "Long-Term Changes to the Microbiome, Blood Lipid Profiles and IL-6 in Female and Male Swedish Patients in Response to Bariatric Roux-en-Y Gastric Bypass"

_nutrients, 2024, doi:10.3390/nu16040498_

Round 1
Reviewer 1 Report
Comments and Suggestions for Authors
Thanks the Editors for the opportunity to review this study. It is a very interesting paper but a few things need clarification.
1. What was the mean BMI of patients before surgery? And what were the qualification criteria (why BMI <35?)? Where they only primary surgery or also revisional? Have the authors disqualified any patients from the study? F.e. intestinal diseases? Please write inclusion and exclusion criteria
2. What was the follow-up rate? All of the patients were examined 6 years after the surgery? Do you have data on individual year after the surgery? In my opinion it would be interesting if the authors will provide results after one year after surgery, when the diet and lifestyle of the patients is rather similar
Author Response
Dear Sir/Madam,
Thank you so much for manuscript reviewing and valuable comments.
Please, find the answers to your questions and comments in the attached file.
Kind regards,
Olena Prykhodko

Reviewer 2 Report
Comments and Suggestions for Authors
I thank the authors for their efforts conductin this research.The article under review presents a compelling exploration into the long-term changes to the microbiome, blood lipids profiles 2 and IL-6 in SWEDISH patients in response to bariatric Roux-en-Y gastric bypass.
Major comments
1. While the study's findings and discussion offer valuable insights into the topic, I find myself concerned about the lack of detailed information regarding some of the methods employed in the study:
a. Blood collection, and storage:
Unfortunately, the authors did not provide sufficient information on the procedures used, making it challenging to evaluate the robustness of the experimental design.
It is imperative that the authors address this oversight by offering a comprehensive account of the blood collection process, including details such as sample size, collection protocols, and any potential biases introduced during this phase. Additionally, information on the storage conditions of blood samples and when exactly it was analysed is crucial, as it directly impacts the stability and integrity of biomarkers.
b. Biomarker analysis:
Thorough elucidation of the biomarker analysis methods is warranted. Readers need to understand the techniques employed, any calibration procedures, and the reliability and precision of the chosen assays. This information is pivotal for the scientific community to contextualize the findings and consider the study's broader implications.
I strongly recommend that the authors revise the manuscript to include a detailed methodology section, ensuring that all aspects of blood collection, storage, and biomarker analysis are explicitly outlined.
2. The authors reported a six-year follow-up period for the patients in their study, a duration that allows for a comprehensive assessment of the long-term effects of bariatric surgery. However, a critical aspect that requires clarification pertains to the timing of the study tests and biomarker analyses within this extended follow-up period.
Understanding when these assessments were conducted is crucial for interpreting the findings accurately. The dynamics of weight loss and metabolic changes following bariatric surgery are known to evolve over time, and fluctuations in weight are common, especially in the years following the procedure. As it is acknowledged, many bariatric surgery patients tend to experience weight regain several years post-surgery, which can significantly impact metabolic parameters and biomarker levels.
Without explicit information regarding the temporal alignment of the study tests and biomarker analyses, there is an inherent challenge in attributing observed outcomes solely to the surgery. The lack of this temporal context introduces ambiguity and limit the study's reliability, as it becomes challenging to distinguish between the immediate postoperative effects and the potential influence of long-term weight changes on the measured biomarkers.
To enhance the robustness of the study, it is recommended that the authors provide a detailed timeline of when the study assessments were performed in relation to the patients' postoperative years.
3. In section 3.1.1. Effect on Body Weight and Quality of Life
a. The authors noted a substantial alteration in body weight following the surgical intervention. However, it would greatly enhance the clarity of the study if the authors could provide additional details, particularly the percentage reduction in weight achieved by the patients.
b. Furthermore, clarification regarding whether this reduction was attained immediately post-surgery, gradually over the six-year follow-up period, and if it was consistently maintained throughout the entire study duration would be helpful in the interpretation of the findings.
Minor comments:
1. Kindly review the manuscript for any minor English errors.
2. Please review the footer of Table 1 to ensure that the abbreviations align with the text before it.
Comments on the Quality of English LanguageThe manuscript need minor revision for english mistakes
Author Response
Dear Sir/Madam,
Thank you so much for manuscript reviewing and valuable comments.
Please, find our response to your comments and questions in the attached file.
Kind regards,
/Olena Prykhodko
